# Single Particle Approaches to Plasmon-Driven Catalysis

**DOI:** 10.3390/nano10122377

**Published:** 2020-11-29

**Authors:** Ruben F. Hamans, Rifat Kamarudheen, Andrea Baldi

**Affiliations:** 1Dutch Institute for Fundamental Energy Research (DIFFER), De Zaale 20, 5612 AJ Eindhoven, The Netherlands; r.f.hamans@vu.nl (R.F.H.); r.kamarudheen@differ.nl (R.K.); 2Department of Physics and Astronomy, Vrije Universiteit Amsterdam, De Boelelaan 1081, 1081 HV Amsterdam, The Netherlands

**Keywords:** plasmonics, heterogeneous catalysis, photocatalysis, nanoparticles, single molecule localization, super-resolution microscopy, surface-enhanced Raman spectroscopy

## Abstract

Plasmonic nanoparticles have recently emerged as a promising platform for photocatalysis thanks to their ability to efficiently harvest and convert light into highly energetic charge carriers and heat. The catalytic properties of metallic nanoparticles, however, are typically measured in ensemble experiments. These measurements, while providing statistically significant information, often mask the intrinsic heterogeneity of the catalyst particles and their individual dynamic behavior. For this reason, single particle approaches are now emerging as a powerful tool to unveil the structure-function relationship of plasmonic nanocatalysts. In this Perspective, we highlight two such techniques based on far-field optical microscopy: surface-enhanced Raman spectroscopy and super-resolution fluorescence microscopy. We first discuss their working principles and then show how they are applied to the in-situ study of catalysis and photocatalysis on single plasmonic nanoparticles. To conclude, we provide our vision on how these techniques can be further applied to tackle current open questions in the field of plasmonic chemistry.

## 1. Introduction

Localized surface plasmon resonances (LSPRs) arise from the coherent oscillation of free electrons upon illumination of metal nanoparticles. These resonances give rise to very large absorption and scattering cross sections, making plasmonic nanoparticles suitable for light harvesting applications such as photocatalysis [1,2,3,4,5,6], solar fuels generation [7,8,9], and integration in photonic devices [10,11,12]. For example, LSPRs have been used to increase the selectivity of propylene epoxidation [13], carbon dioxide reduction [14,15,16], and to increase the rate of H${}_{2}$ dissociation [17] and ammonia decomposition [18]. Despite these demonstrations of enhanced catalytic selectivity and activity, a detailed understanding of the underlying mechanism is still lacking [18,19,20,21,22,23,24,25,26,27,28].

Unraveling the working mechanism of plasmonic catalysts can drive their rational design for specific applications. These studies are often performed at the ensemble level, resulting in measurements that are averaged over many particles and many active sites. The catalytic properties of metal nanoparticles, however, are intrinsically heterogeneous because of multiple factors: (i) even monodisperse nanoparticle ensembles present different reaction pathways for the same catalytic conversion [29], (ii) catalytic activity depends on the available surface facets [30] and the presence of defects [31,32], both of which can vary from particle to particle, and (iii) the structure of a catalyst changes during operation, resulting in spatiotemporal variations in catalytic activity [33,34,35]. Furthermore, in ensemble experiments it is typically difficult to characterize the weak signature of reaction intermediates that are crucial to unveil the reaction pathway [36].

To address these challenges, several techniques have emerged, allowing the study of the spatial and temporal heterogeneity of individual plasmonic nanoparticles and their photocatalytic reaction products under in-situ conditions [37]. Dark-field optical microscopy [38] and transmission electron microscopy (TEM) [33,35] have been used to follow the evolution of the optical and structural properties of active plasmonic photocatalysts with nanoscale spatial resolution. Mass spectrometry [39,40], infrared microscopy [37,39], and X-ray spectroscopy [41], on the other hand, while blind to the properties of the catalytic particles, allow the detection of the reaction products during the photocatalytic process. Furthermore, near-field techniques, such as infrared scanning near-field optical microscopy [42,43], tip-enhanced Raman spectroscopy [44,45], and scanning electrochemical microscopy [46,47] allow the interrogation of individual catalytic nanoparticles and obtain information on both their structure and catalytic performance. These techniques, however, typically have limited temporal (dark-field optical microscopy) or spatial (mass spectrometry) resolution and sensitivity, or require the use of expensive specialized equipment (TEM, infrared microscopy, near-field microscopy, and X-ray spectroscopy).

In this perspective, we highlight two experimental techniques that can be implemented in plasmonics research labs using conventional far-field optical microscopes and that are capable of detecting reaction products and intermediates both in real time and with nanometer spatial resolution: surface-enhanced Raman spectroscopy (SERS) and super-resolution fluorescence microscopy. SERS gives information on which intermediates and reaction products are present on the nanoparticle surface, but lacks sub-particle spatial resolution. Conversely, super-resolution fluorescence microscopy is able to characterize chemical reactions with nanometer resolution, but does not give any chemical information. We first describe the optical properties of metallic nanoparticles and the mechanisms by which they can enhance the rate of a chemical reaction. We then discuss the working principles of the two techniques and highlight selected articles that describe key observations made using these techniques in the context of plasmonic chemistry. Finally, we propose how these techniques can be further applied to study current open questions in plasmon-driven catalysis. For comprehensive reviews on SERS and super-resolution fluorescence microscopy in the context of catalysis, we refer to already existing literature [48,49,50,51,52,53].

## 2. Optical Properties of Metallic Nanoparticles

Let us consider the simple case of a metallic nanoparticle immersed in a harmonically oscillating electric field. The easiest case that can be solved analytically is that of a spherical nanoparticle with a radius much smaller than the oscillation wavelength. In this quasi-static approximation, the phase of the electric field is constant over the entire volume of the nanoparticle, and the problem can be simplified into that of a particle in an electrostatic field. This approximation allows us to rewrite Maxwell’s equations into the Laplace equation, which, in the geometry considered here, allows us to define the polarizability α of the sphere as [54]:(1)$$\alpha =4\pi R^{3}\frac{\epsilon -\epsilon_{m}}{\epsilon +2\epsilon_{m}},$$
where *R* is radius of the sphere, ϵ is the wavelength-dependent complex permittivity of the sphere, and $\epsilon_{m}$ is the permittivity of the surrounding medium. The scattering ($\sigma_{\mathrm{sca}}$) and absorption ($\sigma_{\mathrm{abs}}$) cross sections can be calculated from the polarizability using [54]:(2)$$\sigma_{\mathrm{sca}}=\frac{k^{4}}{6\pi }{\left|\alpha \right|}^{2},$$
(3)σabs=kIm[α],
where k=2π/λ is the wavevector. From the above equations it is clear that the particle polarizability, and therefore its scattering and absorption cross sections, diverge when $\epsilon =-2\epsilon_{m}$. For typical plasmonic metals such as gold and silver in vacuum ($\epsilon_{m}=1$) or water ($\epsilon_{m}=1.777$) this condition is satisfied in the visible part of the electromagnetic spectrum (Figure 1a). The magnitude of α remains finite due to the non-vanishing imaginary part of the permittivity ϵ, which accounts for losses in the metal (Figure 1b).

For larger spheres the electric field phase variation inside the nanoparticle needs to be taken into account. If we consider a plane wave impinging on a spherical particle of arbitrary size in a homogeneous medium, this problem can still be treated analytically thanks to its spherical symmetry, resulting in the so-called Mie theory [54,57]. The analytical solution to Mie theory yields the electric fields inside and outside the particle, as well as its scattering and absorption cross sections. As shown in Figure 1c, the resulting extinction cross section ($\sigma_{\mathrm{ext}}=\sigma_{\mathrm{sca}}+\sigma_{\mathrm{abs}}$) can be many times larger than the geometrical one, demonstrating the ability of plasmonic nanoparticles to focus far-field radiation into sub-wavelength volumes.

In realistic experimental conditions, however, photocatalytic nanoparticles are typically not perfectly spherical and are not embedded in a homogeneous medium, for example when a support is used. In fact, it has been shown that highly anisotropic particles with sharp tips, such as nanostars and octopods, show enhanced catalytic activity when compared to spheres [58,59,60]. To calculate the electric fields and the scattering and absorption cross sections for arbitrary geometries, Maxwell’s equations need to be solved numerically, for example using a finite-difference time-domain method or a finite elements method.

The resonance condition $\epsilon =-2\epsilon_{m}$ shows that the spectral position of the LSPR can in principle be tuned by changing the permittivity of the surrounding medium (Figure 1a) or the composition of the particle (Figure 1c). In reality, however, these parameters cannot be set arbitrarily, as photocatalytic experiments are usually performed either in reactive gas environments or in aqueous solutions and the composition of the particle cannot be changed at will.

Nevertheless, for SERS the LSPR wavelength needs to be matched to the laser wavelength that is used to probe the molecular vibrations and for super-resolution fluorescence microscopy the LSPR wavelength needs to be spectrally separated from the emission of the reaction products to avoid mislocalization effects [61,62]. This necessary spectral tunability of the LSPR is typically achieved by synthesizing anisotropic particles such as rods, prisms, or cubes [63,64]. For example, for plasmonic nanorods an increasing aspect ratio results in a longer LSPR wavelength of their dominant resonance (Figure 1d).

The radiative decay of an LSPR (scattering) gives rise to intense electromagnetic fields on the nanoparticle surface, which are typically known as near-fields. The magnitude of these scattered fields can be amplified to values several orders of magnitude higher than the incoming electric field, depending on the nanoparticle morphology and composition. This phenomenon is responsible, for example, for the amplified Raman signal on plasmonic nanostructures, as will be discussed further in Section 3. The non-radiative decay of an LSPR (absorption) results in the formation of highly energetic non-equilibrium charge carriers which can drive redox reactions [2]. These hot electrons and holes are typically extremely short-lived, due to the high charge carrier density of metals [65,66]. When these carriers are not harvested, they dissipate their energy via electron-electron and electron-phonon scattering events [67], thereby heating up the particle and, eventually, its surrounding environment (Figure 1e) [2]. The latter photothermal process can also accelerate temperature-dependent chemical reactions according to the Arrhenius equation [68].

## 3. Surface-Enhanced Raman Spectroscopy

The radiative decay of plasmon resonances can result in a strong amplification of the electromagnetic fields at the surface of metal nanoparticles. Nanostructures possessing sharp corners and tips with low radii of curvature, or metal structures separated by nanoscale gaps such as dimers, can be utilized to amplify the magnitude of these plasmonic near-fields. For example, at resonant illumination, 75 nm Ag nanocube dimers separated by a 1 nm gap can sustain hot spots with electric field intensities up to 6 orders of magnitude higher than the incident radiation (Figure 2a) [69]. These electromagnetic hotspots can be exploited for a wide variety of applications ranging from driving photo-sensitive reactions [20,70,71,72], to enhancing efficiencies in photovoltaic devices [73], amplifying photoluminescence in semiconductors [74], and characterizing catalytic reactions occurring on the nanoparticle surfaces [53,75,76].

In particular, the Raman vibrational signals of molecules adsorbed on the nanoparticle can be strongly enhanced by the plasmonic near-fields, as these Raman signals are approximately proportional to the 4th power of the local electromagnetic field [77]. Such strong field dependence has its origin in the relatively broadband features of typical plasmon resonances (Figure 1c). The dipole-dipole interaction of the adsorbed molecule with the plasmonic near-fields, in fact, can enhance both light absorption at the Raman pump frequency as well as light emission at the slightly-shifted Stokes and Anti-Stokes scattering frequencies [77,78]. Previously, near-field enhancements have been used to amplify the Raman signals of molecules up to $10^{10}$ times with respect to the signal in the absence of a plasmonic nanostructure [79]. This technique, referred to as surface-enhanced Raman spectroscopy (SERS), can provide improved spatial, temporal, and spectral resolution compared to normal Raman characterization of molecules [53,80,81,82,83]. Since the plasmonic near-fields are confined to a few nanometers from the nanoparticle surface, SERS selectively provides information of molecules adsorbed to the nanoparticle, hence minimizing background noise and pushing detection limits down to single molecules [84,85,86,87,88]. The large SERS signal enhancement has also enabled measurements with lower acquisition times, of the order of a few milliseconds, while maintaining a high signal-to-noise ratio, thereby allowing to monitor chemical and diffusion dynamics of surface adsorbates [84,89].

In the context of plasmon-driven catalysis, a single laser can be used to both drive catalytic reactions as well as characterize their SERS signals, thereby eliminating the need of complicated setups [90,91,92]. Below, we briefly describe the typical experimental setup of SERS measurements followed by a few examples of how this technique has been used to gain mechanistic understanding of plasmon-driven reactions.

Typically, single particle SERS measurements are performed by depositing dilute suspensions of colloidal plasmonic nanoparticles on transparent substrates to allow the interrogation of individual nanoparticles [46,53,93]. Alternatively, nanoparticles fabricated using a top-down approach such as focussed ion beam milling, nano-imprint lithography, electron-beam lithography, or hole-mask colloidal lithography can also be employed [45,94]. These nanoparticles are then illuminated with a focused laser beam, typically through an objective lens [46]. The laser photon energy is chosen to overlap with the LSPR, so as to obtain large near-field enhancements and thus high SERS signal-to-noise ratios.

The SERS detection of molecules on single metal nanoparticles can be performed in both transmission and reflection microscope configurations (Figure 2b). Since SERS sensitivity is the highest at the surface of the nanoparticle, it is necessary to properly identify its spatial position before irradiating it with a focused laser. As such, in both these geometries, first, a dark-field image of the substrate is typically obtained by focusing a broadband light source at large incident angles using an objective possessing a high numerical aperture (NA) [95]. The scattered light is then collected with an objective with a lower numerical aperture. Such an illumination geometry results in a dark background, where the plasmonic nanoparticles can be identified as small, diffraction-limited, bright scattering spots (Figure 2c). Once the nanoparticle position on the substrate is determined, a focused laser beam is used to excite the Raman vibrational modes of molecules adsorbed on its surface and the scattered light is guided to a CCD camera coupled with a grating. Often, a notch filter is kept in the scattering pathway, in order to exclude any laser damage to the spectrometer. The analyte of interest can either be adsorbed before depositing the nanoparticles on the substrate or it can be introduced in-situ using a flow cell configuration.

SERS measurements on single nanoparticles can provide valuable catalytic information such as the identification of reaction intermediates and the reaction kinetics [51,53,92,96]. Initial studies in this field focused on the model reduction of p-nitrobenzenethiol (NBT) to p-aminobenzenethiol (ABT) or p,p′-dimercaptoazobenzene (DMAB) catalyzed by metal nanoparticles. The advantage of employing such model reactions is that the reactants and products display strong Raman signals, which facilitate easy characterization of the process kinetics [45]. The absence of many competitive side reactions and the ability to operate at ambient conditions makes it a relatively easy system to study [97]. The strong affinity of the thiol functional group to the metal surface also ensure that the measured SERS signal originates exclusively from molecules adsorbed to the nanoparticles.

The rate of these model reactions can be enhanced by plasmonic hot electrons. For example, Kang et al. employed single Ag microparticles (∼2 μm diameter) with rough surfaces as SERS substrates to analyze the plasmon-driven reductive dimerization of NBT to DMAB (Figure 2d) [80]. Nanoparticles with rough surfaces have electromagnetic field enhancements that are significantly higher than their smooth counterparts and can therefore act as excellent Raman substrates. The plasmonic substrates were functionalized with NBT and then deposited on a silicon wafer, followed by laser illumination under atmospheric conditions. Time-resolved SERS spectra of the reduction reaction revealed the disappearance of the ν(NO${}_{2}$,sym) peak of NBT (1335 cm${}^{-1})$ as the reaction proceeded, along with an increase in the ν(N=N,sym) peak of DMAB (1440 cm${}^{-1})$. To elucidate the influence of the illumination parameters on the reaction rate, SERS characterization was performed at various illumination powers using 532 nm and 633 nm continuous wave lasers. Under 532 nm illumination the reaction kinetics were significantly faster than under 633 nm illumination at constant laser power. This difference in the reactivity was attributed to the change in the absorbed power in the nanoparticle and the subsequent change in the number of hot charge carriers ejected. Strikingly, aminobenzenthiol (ABT) which has been reported as a reduction product of NBT was not observed in these experiments. To gain mechanistic insights in to the selectivity of the NBT reduction, the authors performed additional experiments under controlled environments. They observed that in the presence of H${}_{2}$O or H${}_{2}$, the DMAB products are further reduced into ABT [98]. Such experiments show how SERS can be used to follow reaction kinetics of catalytic reactions occuring on a single nanoparticle photocatalyst with controlled or predetermined geometry and surface sites.

Often, the exact plasmon-activation process behind photochemical reactions is challenging to identify, as the different competing mechanisms occur at ultra short timescales [22,28]. Photothermal effects have been used as an alternative explanation to many of the reported plasmon-driven chemical reactions [27]. SERS measurements offer an opportunity to quantitatively assess these thermal effects in both illuminated ensemble and single nanoparticle systems [99,100,101]. By comparing the Anti-Stokes $\left(I_{AS}\right)$ and Stokes $\left(I_{S}\right)$ intensity modes of the adsorbate, the vibrational temperature can be quantified using the relationship [101]:(4)$$\frac{I_{AS}}{I_{S}}=A\left[\frac{\tau \sigma_{s}^{{}^{\prime }}I}{h\nu_{L}}+\exp\frac{-h\nu_{vib}}{k_{b}T}\right]$$
where, *A* is a constant that takes into account the electric field enhancement at the nanoparticle surface and the Raman cross-section of the molecule, τ is the lifetime of the vibrational excited state, $\sigma_{s}^{{}^{\prime }}$ is the Raman cross-section at the Stokes signal, *I* is the laser intensity, ν is the frequency, $k_{b}$ is the Boltzmann constant, and *T* is the temperature experienced by the molecule. The subscripts *L* and vib represent laser and Raman vibrational mode respectively.

For example, Wu et al. ruled out the contribution of photothermal effects in the plasmon-driven electron transfer to [6,6]-phenyl-C${}_{61}$-butyric acid methyl ester by extracting the temperature of their Au nanoparticle suspension using Equation (Equation 4) [100]. In this study, the temperature increase at various illumination intensities was found to be negligible, thereby confirming the electronic origin of plasmonic effects [100]. Similarly, Pozzi et al. investigated the temperature experienced by Rhodamine 6G molecules adsorbed onto single Ag nanoparticle aggregates using SERS measurements [101]. In their report, they highlighted that accurate temperature measurements using Equation (Equation 4) should be accompanied by careful estimation of the different electric field enhancements and molecular Raman cross-sections at the Stokes and Anti-Stokes frequencies, both of which contribute to the value of the constant *A* [101].

SERS measurements on single nanoparticles have also been used to provide mechanistic insights into plasmon-driven CO${}_{2}$ reduction (Figure 2e) [93]. Kumari et al. chose Ag nanoparticles as SERS substrates for this purpose, as they typically display larger near-field enhancements compared to metals such as Au and Cu. In their experiments, a focused 514.5 nm laser was used to excite the SERS spectra of adsorbed reactant and product molecules on 60 nm Ag nanosphere aggregates dispersed on a glass substrate. Time-resolved SERS spectra displayed the generation of carbon monoxide, hydrogen, formic acid, formaldehyde, methanol, and carbonate molecules on the nanoparticle surface. The blinking nature of the SERS signals in this study suggests that only a single molecule is present over the nanoparticle surface at a time. Interestingly, the authors observed the generation of a surface-adsorbed hydrocarboxyl radical HOCO* during CO${}_{2}$ reduction. Using density functional theory calculations, this intermediate was found to be crucial in determining the selectivity of the photoreduced products. Thanks to the extreme field confinement on plasmonic nanoparticles, single particle studies can detect the stochastic adsorption and desorption of reactants and products along with the formation of intermediate species at millisecond time resolutions. Such single molecule and real-time detection capability is unachievable using ensemble studies due to signal averaging. Single particle measurements focused on identifying the reaction intermediates will play a pivotal role in elucidating structure-activity relationships of future nanostructured photocatalysts.

Recently, SERS measurements have also been employed to obtain mechanistic insights into plasmon-driven ethylene epoxidation on Ag nanospheres [36]. The time evolution of the Raman signals indicated a correlation between the formation of a graphene layer on the nanoparticle surface and the formation of ethylene oxidation species. Further experiments using a graphene coated Ag nanoparticle, along with DFT calculations, confirmed that the in-situ formed graphene, rather than the Ag nanoparticle, is the catalyst for ethylene epoxidation. Such mechanistic insight could not have been obtained with ensemble measurements, in which it would have been impossible to distinguish the narrow signature of a graphene nanocrystal among the broad Raman spectra of mixed carbonaceous species typically obtained upon ensemble averaging. Understanding the role of graphene nanocrystals in ethylene epoxidation could inform the development of industrial photocatalysts, as ethylene oxide is an important feedstock in the synthesis of solvents, plastics, and other organic chemicals.

Choi et al. conclusively demonstrated single molecule plasmon-driven conversion of NBT to ABT, by sandwiching the reactant molecules between a Au thin film and a Ag NP (Figure 2f) [84]. Such geometries are commonly referred to as nanoparticle on mirror (NPoM) or as particle over surface. The nanoscale gap between the nanospheres and the film generates a SERS enhancement factor of 10${}^{8}$ that varies by less than an order of magnitude between individual particles. Such reproducibility in the SERS geometries can be attributed to the fixed length of the spacer molecule, in this case NBT. The large signal-to-noise ratio of the SERS spectra allowed the authors to perform measurements at millisecond time scales, thereby accurately quantifying the reaction kinetics. In this study, the SERS measurements displayed discrete jumps in the DMAB intermediate signal while the NBT reactant signal continuously decayed. The discrete spectral jumps indicate the conversion of a single molecule, which was further corroborated with a kinetic Monte Carlo model accounting for the spatial variations in the near-field enhancements and the Raman cross sections of the molecules.

The authors also used the kinetic information to obtain mechanistic insights into the reduction of NBT. Two main reaction mechanisms had been previously proposed for the conversion of NBT to ABT, namely direct and indirect (Figure 2g). The direct mechanism generated ABT via the formation of dihydroxyaminobenzenethiol (DHABT), nitrosobenzenethiol (NSBT), and hydroxylaminobenzenethiol (HABT) intermediates, while the indirect path involved condensation of DHABT and HABT to form DMAB and its subsequent reduction to ABT. By analysing the rate of NBT consumption and the rate of DMAB formation, the authors pointed out that only less than 10% of NBT molecules undergo reduction via the indirect path (Figure 2h). Such deep insights into the reaction mechanism reveals the power of SERS measurements to identify spectro-temporal behavior in single molecule photocatalytic experiments, without the addition of any label molecules.

## 4. Super-Resolution Fluorescence Microscopy

Optical fluorescence microscopy is a powerful imaging technique typically used for the structural characterization of biological samples. However, due to diffraction its spatial resolution remains limited to ∼λ/2NA [102], corresponding to ∼200 nm for visible wavelengths λ and objectives with high numerical aperture (NA≈1.4). This limitation in resolution can be overcome when imaging the fluorescence of a single molecule. Under the condition of a single fluorophore, in fact, the measured emission pattern can be fit to a two-dimensional Gaussian, where the center of the Gaussian is the position of the molecule [103]. The localization precision $\sigma_{\mathrm{loc}}$ of this position can be calculated using [104]:(5)$$\sigma_{\mathrm{loc}}^{2}=\frac{\sigma_{a}^{2}}{N}\left(\frac{16}{9}+\frac{8\pi \sigma_{a}^{2}b^{2}}{Na^{2}}\right),$$
(6)$$\sigma_{a}^{2}=\sigma^{2}+a^{2}/12,$$
where σ is the standard deviation of the Gaussian, *a* is the size of a camera pixel, *N* is the number of detected photons, and *b* is the standard deviation in the background noise. Under the experimental conditions discussed in this Perspective, $\sigma_{\mathrm{loc}}$ is typically between 10 and 30 nm, which is commensurate with the typical size of plasmonic photocatalysts.

Although originally developed to image sub-diffraction limited features in biological samples [105,106,107], the use of super-resolution microscopy has recently been extended to a variety of research fields, from the optimization of nanophotonic devices [108,109,110], to the characterization of nanomaterials [111]. Recently, single molecule localization has been extended to the mapping of heterogeneous chemical reactions with nanoscale spatial resolution and single molecule turnover accuracy. For example, super-resolution microscopy has been used to study so-called fluorogenic reactions, catalytic conversions producing a fluorescent reaction product [112]. In a typical experiment, catalysts are dropcasted on a glass substrate, which is built into a flowcell that allows a continuous supply of reactants. The catalysts are dropcasted at a sufficiently low concentration, such that individual particles can easily be resolved spatially. The sample is then illuminated with a laser that excites the fluorescence of the reaction products. To minimize background signal, this illumination is usually done in a total internal reflection fluorescence (TIRF) configuration [113], either using the same high NA objective that is used to capture the fluorescence (Figure 3a), or using a prism (Figure 3b).

Illumination through the objective (Figure 3a) requires the NA to be higher than the critical angle for TIRF, which implies NA > 1.333, assuming the surrounding medium is water. In practice, an NA of ∼1.45 is typically used, as otherwise only a very small part of the peripheral area of the lens can be used for TIRF illumination, making the alignment procedure challenging. Despite being more costly, an objective with a higher NA also results in a superior collection efficiency thanks to its high acceptance angle. Since the localization precision scales with $1/\sqrt{N}$ (see Equation (Equation 5)), a higher collection efficiency improves the localization precision. Furthermore, illumination through the objective allows for easy access to the sample from the top. Illumination through a prism (Figure 3b) has a lower collection efficiency due to the typically lower NA objectives used and can suffer from light scattering by impurities in the liquid. On the other hand, this illumination method is less costly, as it does not require extremely high NA, and suffers less from fluorescence quenching, as the nanoparticle catalyst is not in the detection pathway.

Both illumination methods are widefield, which allows many particles to be measured simultaneously (Figure 3c). Since the typical integration time that is needed to acquire enough photons to localize a single fluorophore is between 10 and 100 ms, only fluorescent molecules that are adsorbed on the catalyst are detected. Fluorescent species that are free in solution, in fact, move too fast due to Brownian motion and only contribute to the background signal. In fluorogenic reactions, upon product formation a fluorescent burst is detected, until the product molecule desorbs from the catalyst surface and diffuses away (Figure 3d). Each fluorescent burst (Figure 3e) is fitted to a two-dimensional Gaussian (Figure 3f), which gives the in-plane position of each molecule with nanometer accuracy.

In 2008, Xu et al. reported the real-time imaging of redox catalysis on ∼6 nm gold particles with a time resolution up to 30 ms [29]. The reaction under study was the gold-catalyzed reduction of the nonfluorescent molecule resazurin to the highly fluorescent molecule resorufin in the presence of NH${}_{2}$OH as a reducing agent. Despite the fairly narrow size distribution of the gold particles (6.0 ± 1.7 nm), the study revealed a large heterogeneity in turnover rate. Furthermore, the authors showed that within a single batch of nanoparticles different reaction pathways exist for the desorption of the resorufin molecules from the gold surface. While most particles preferred product desorption assisted by a reactant binding step, some preferred a direct desorption of the product molecule resorufin, and some others showed no preference between the two mechanisms. This diverse behavior demonstrates the heterogeneity in catalytic properties of metal nanoparticles and highlights the importance of single particle approaches to unveil their structure-function relationship. This study was later extended to particles of varying sizes up to ∼14 nm, which showed that larger particles have a lower turnover rate when normalized to the surface area, are less selective between the different desorption mechanisms, and are less prone to restructuring [115].

In both of these studies the size of the catalyst was smaller than the spatial resolution typically achievable with super-resolution microscopy, which hinders the imaging of spatial heterogeneities within a single catalytic nanoparticle. For larger catalysts, however, their activity can be resolved spatially. Figure 3g shows a two-dimensional histogram of the detected resorufin molecules catalytically produced at the surface of a single Au nanorod [31]. By segmenting individual nanorods and determining the specific turnover rate for each segment, it was found that the nanorod tips generally have higher reactivity than their cores. This difference in reactivity can be attributed to the presence of more corner and edge sites at the nanorod tips, which are typically more reactive. Interestingly, when only analyzing the reactivity of the nanorod core, i.e., without the tips, the authors found a specific turnover rate that decreased with increasing distance from the center, even though the core is made up of the same facets. This reactivity gradient can be attributed to the growth mechanism of the nanorod. During synthesis, the growth rate of the nanorod decreases linearly with increasing length [116]. Since faster colloidal growth is typically accompanied by a higher density of superficial defects, a gradient in defect density develops, resulting in higher reactivity at the nanorod center [31]. These sub-particle catalysis maps demonstrate that, besides the types of surface facets [30], defects play a large role in the catalytic activity of metal nanoparticles [31,32].

Additionally, sub-particle spatial resolution allows the visualization of spatiotemporal variations in catalytic activity due to surface restructuring. Measurements of the catalytic activity of triangular Au nanoplates showed that the spatial distribution of catalytic events slowly varies over timescales of several hours [34]. Although the reactivity between different corners of the same nanoplate was found to initially be heterogeneous, over time all corners converged to a similar reactivity. This slow loss of heterogeneity was attributed to the dynamic restructuring of the nanoplate corners from sharp to blunt, as also confirmed with ex-situ TEM [34].

Lastly, spatiotemporally-resolved single molecule catalysis imaging has recently been used to study the correlation between events on a single catalyst, and between events on different catalysts [117]. The authors found that catalytic reactions on individual Pd or Au nanoparticles were correlated over lengths scales of ∼100 nm, probably via the transport of positively charged holes over the catalyst surface. The events between different nanoparticles was also found to be correlated over many micrometers via the molecular diffusion of negatively charged reaction products. Both the influence of defects and of dynamic restructuring on catalytic activity and the correlation between different events would have remained hidden in ensemble or diffraction-limited measurements.

Although the fluorogenic catalytic reactions used in these seminal studies typically also happen in the dark, their rates can be enhanced under laser excitation [114,118,119]. Measurements on CdS semiconductor nanorods decorated with Au nanoparticle co-catalysts at their tips showed that plasmon excitation (below bandgap) and semiconductor excitation (above bandgap) both enhance the rate of Amplex Red oxidation to resorufin [114]. Photogenerated electrons promote desorption of the negatively charged resorufin molecule due electrostatic force. Therefore, electron-driven turnovers are characterized by short fluorescent bursts. Conversely, photogenerated holes enhance adsorption of resorufin and, therefore, increase the length of the fluorescent bursts. On the CdS-Au nanostructures two distinct burst lengths were observed, which allowed the authors to distinguish between electron-driven and hole-driven reactions. When the plasmon resonance of the Au nanoparticles is excited (below bandgap) hot electrons and holes are generated within the Au. The photogenerated electron is injected into the conduction band of the semiconductor due to the Schottky barrier that is formed at the CdS-Au interface, and the photogenerated hole remains in the Au. This charge separation is also observed in the super-resolution catalysis maps (Figure 3h), with the hole-driven reactions (long bursts) taking place on the edges of the rod where the Au nanoparticles are located, and the electron-driven reactions (short bursts) taking place a few tens of nanometers inward on the CdS nanorod. For above bandgap excitation, charge carriers are generated in the CdS nanorod. The photogenerated electrons transfer to the Au nanoparticles, whereas the holes remain in the nanorod. This charge separation is again reflected in the super-resolution catalysis maps (Figure 3h), with the hole-driven reactions now taking place all over the nanorod and the electron-driven reactions on the Au nanoparticles [114].

On dimers of Au nanorods with nanoscale gaps, the turnover rate of resazurin to resorufin reduction was also found to be dependent on the incident laser power [118]. The turnover rate scaled quadratically with laser power and, therefore, the enhancement can be attributed to the presence of two photoexcited species on the nanoparticle surface. However, the incident laser photoexcites both the reactant resazurin and the plasmon resonance of the catalysts, preventing a straightforward distinction between near-field effects (photoexcitation of the reactants) and hot charge carrier effects [118].

Another approach of characterizing the influence of plasmon excitation on the catalytic reactivity of nanoparticles is by measuring the change in the single particle activation barrier of the fluorogenic reaction when the plasmon resonance is excited. According to the Arrhenius equation, both the product formation and the product desorption rates, ${\langle \tau_{off}\rangle }^{-1}$ and ${\langle \tau_{on}\rangle }^{-1}$, respectively, depend on temperature via [120]
(7)$${\langle \tau_{off}\rangle }^{-1}=A_{off}\exp\left[-E_{a,off}/RT\right],$$
(8)$${\langle \tau_{on}\rangle }^{-1}=A_{on}\exp\left[-E_{a,on}/RT\right],$$
where $A_{off}$ and $A_{on}$ are prefactors, $E_{a,off}$ and $E_{a,on}$ are the activation barriers for the product formation and desorption processes, respectively, *R* is the gas constant, and *T* is the temperature. Therefore, by measuring $\langle \tau_{off}\rangle$ and $\langle \tau_{on}\rangle$ for varying reactor temperatures, the single particle activation barriers for the product formation and desorption can be extracted [120,121]. For chemical reactions with an intermediate species, also a distinction can be made between the activation barrier for intermediate formation and for final product formation [119]. Single molecule experiments on the Au nanorod catalyzed oxidation of Amplex Red to resorufin showed that upon plasmon excitation, the activation barrier for intermediate formation decreased, while the activation barriers for product formation and product desorption remained unaltered [119]. This lowering in activation barrier was attributed to the presence of hot charge carriers. Thermal effects were ruled out based on calculations and thermocouple measurements. Furthermore, no change in activation barrier was observed for the resazurin reduction to resorufin, which would be expected if the plasmonic enhancement was purely thermal [119].

One obvious limitation of the super-resolution fluorescence microscopy studies presented so far is that the reaction under study needs to generate a fluorescent molecule. Most biological and chemical processes, however, do not involve fluorescent species. This limitation has recently been overcome using a competition-based technique, in which a single nanoparticle can catalyze two chemical conversions [122]. The first reaction is the reaction of interest and its reactants and products do not fluoresce. The second auxiliary reaction is fluorogenic and can be imaged and localized with nanometer spatial resolution. If both reactions compete for the same surface sites, the reaction of interest suppresses the rate of the fluorogenic reaction. The extent of suppression can be imaged using super-resolution microscopy, thereby giving spatial information on the reaction of interest.

## 5. Outlook

We began this Perspective by introducing the optical properties of metallic nanoparticles. We discussed how the decay of plasmon resonances generates hot charge carriers, near-field enhancements, and elevated nanoparticle surface temperatures, all of which can simultaneously contribute to drive chemical reactions. We have then introduced two far-field optical microscopy techniques, surface-enhanced Raman spectroscopy and super-resolution fluorescence microscopy, and we have highlighted how they can disentangle the activation mechanism of plasmon-driven chemical reactions thanks to their unique temporal and spatial resolutions. In this final outlook section, we provide our perspective on how these techniques can be further used to tackle current open questions in the field of plasmon-driven chemistry.

As we have seen, the resazurin-to-resorufin conversion is often used to benchmark plasmonic photocatalysts at the single particle level. Super-resolution fluorescence microscopy measurements suggest that hot charge carriers are the main driving force behind the increased turnover rate observed under plasmon excitation. However, the reaction product resorufin and the plasmon resonance of the metal nanoparticle are typically excited using the same 532 nm laser. At this wavelength the reactant resazurin is weakly absorbing, which can lead to its photodriven disproportionation to resorufin. This side reaction is enhanced by the strong plasmonic near-fields at the nanoparticle surface and can therefore entirely by-pass any photocatalytic charge transfer process [123]. To decouple plasmon-driven charge transfer effects from the photoexcitation of reactant molecules, it would be desirable to design experiments involving spectrally detuned plasmonic catalysts and two lasers, one for the excitation of the fluorescent molecules and one for the plasmon resonance of the catalyst [118,119].

Furthermore, to confidently eliminate the contribution of plasmonic heating in both SERS and super-resolution measurements, control experiments should be performed where the nanoparticle system is externally heated to the measured temperature [27]. Ideally, in-situ experimental characterization of the temperature should be performed [101,124,125], for example by analyzing the Anti-Stokes and Stokes signals of the SERS spectra. In the absence of experimental measurements, theoretical calculations can give an estimate of the surface temperature of the irradiated nanoparticle [125,126,127,128]. Although such experiments may seem obvious, they are often challenging in practice, especially when studying single nanoparticles using an optical microscope. Under these conditions, in fact, laser irradiation can easily heat the nanoparticle under study to hundreds of degrees Celsius and these elevated temperatures are difficult to reach with conventional thermally-controlled sample holders. The lack of simple thermometric techniques at the nanoscale, along with the challenges in accurately calculating temperature increases under *operando* conditions, make it difficult to effectively assess the temperature at the active site of a plasmonic photocatalyst [127,129].

To establish structure-activity relationships, it is crucial to characterize the morphology of the nanocatalysts. This can be done using dark-field scattering spectroscopy or electron microscopy before, during, and after the catalytic reaction, along with the SERS or super-resolution characterization [38]. Most SERS studies on catalytic reactions are performed on nanoparticle aggregates rather than single particles. Extending SERS measurements to single particles could allow detecting catalytic intermediates with higher signal-to-noise ratios, and establishing more accurate structure-activity relationships. Another strategy to ensure that these measurements are performed on individual structures with a controlled size is by utilizing nanoparticle substrates fabricated using a top-down approach such as electron-beam lithography [45,130,131,132]. An additional advantage of using top-down techniques to deposit plasmonic photocatalysts is that the surface of the nanoparticles is typically free of contaminants or ligands, hence minimizing their potential contribution to the measured catalytic activity.

Lastly, plasmonic enhancements are usually characterized by the single particle turnover rate, given by the number of detected products per second. However, the nanoscale spatial distribution of catalytic events, as unveiled by super-resolution fluorescence microscopy, can also provide valuable information on the underlying activation mechanism [129,133]. Both the plasmonic near-fields and the generation of non-equilibrium charge carriers, in fact, are strongly polarization dependent [67,134], whereas the nanoparticle temperature is expected to be homogeneous due to the high thermal conductivity of the metal [2,133,135].

Studying light-driven reactions on metal photocatalysts in real-time and with single particle spatial resolution is a challenging task that promises to unveil previously inaccessible information on the photocatalyst active sites, the photoactivation mechanism, the molecular intermediates, and the reaction pathways. Here we have presented recent successes and future promises of two far-field optical techniques, that have recently allowed us a closer look on these fundamental nanoscale processes.

## Figures and Tables

**Figure 1 nanomaterials-10-02377-f001:**
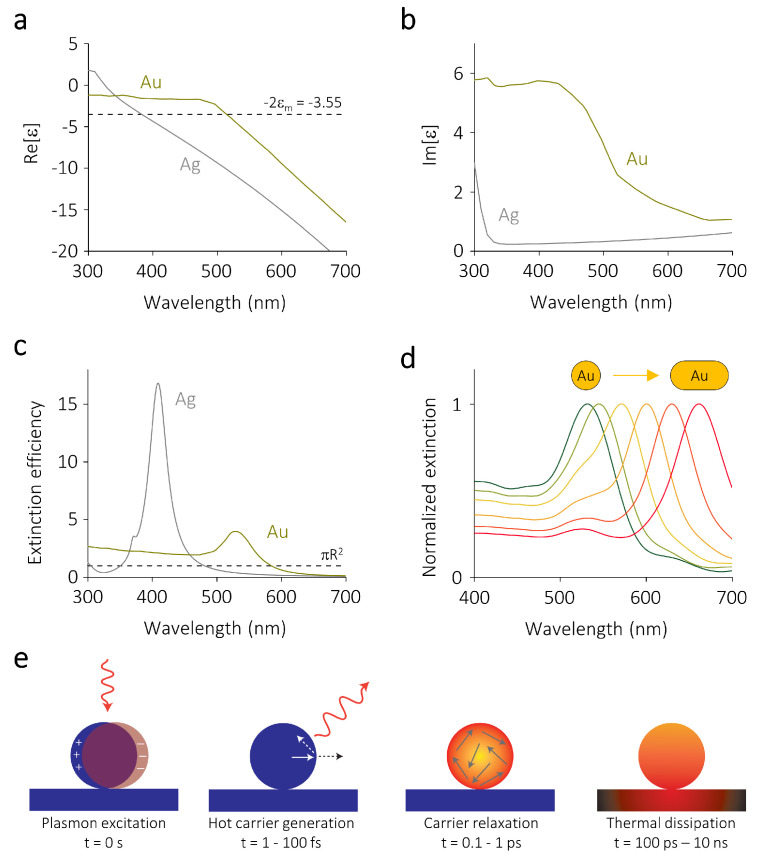
Optical properties of metallic nanoparticles. (**a**) Real and (**b**) imaginary part of the wavelength-dependent permittivities of gold (Au) and silver (Ag). The dotted line in panel (**a**) denotes the resonance condition $\epsilon =-2\epsilon_{m}$ when the surrounding medium is water. The permittivities are taken from the literature [55,56]. (**c**) Extinction cross sections of spherical nanoparticles of gold and silver with radii *R* = 25 nm. The cross sections are calculated using Mie theory and are normalized to the geometrical cross section $\pi R^{2}$, which is denoted by the dashed horizontal line. (**d**) Normalized extinction cross section of a gold sphere of *R* = 25 nm, that is elongated in steps of 10 nm to a gold rod with a total length of 100 nm. The cross sections are calculated using a finite-difference time-domain method. (**e**) Localized surface plasmon resonances (LSPRs) can decay radiatively (scattering) into photons or non-radiatively (absorption) into non-equilibrium charge carriers. These carriers relax via electron-electron scattering, followed by electron-phonon scattering, which heats up the nanoparticle and eventually also the surrounding medium. Panel (**e**) reproduced with permission of [2]. Copyright Nature Publishing Group, 2015.

**Figure 2 nanomaterials-10-02377-f002:**
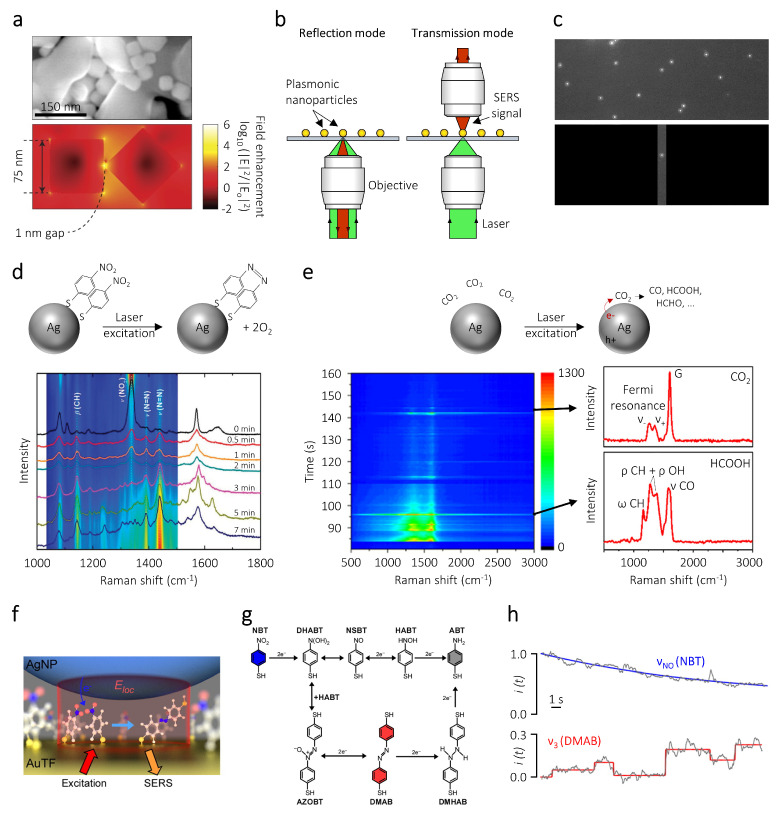
Surface-enhanced Raman spectroscopy (SERS). (**a**) (**top**) Scanning electron micrograph of silver nanocubes supported on alumina particles. The cubes arrange randomly, thereby generating geometries that can provide high electric fields. (**bottom**) Finite-difference time-domain simulation of the spatial distribution of the electric fields of 75 nm Ag nanocubes separated by 1 nm gap. The calculation is performed at the resonance wavelength of 500 nm. (**b**) Schematic illustration of SERS measurements over single nanoparticles in reflection and transmission mode. (**c**) (**top**) Grayscale dark-field image of several single plasmonic nanoparticles deposited on a transparent substrate. (**bottom**) Dark-field image of an individual nanoparticle obtained by closing the parallel slit located at the entrance of the monochromator, corresponding to the image plane outside the microscope. (**d**) (**top**) Schematic representation of the plasmon-driven reduction of p-nitrobenzenethiol (NBT) to p,p′-dimercaptoazobenzene (DMAB). (**bottom**) Time-resolved SERS measurement of the NBT reduction. (**e**) (**top**) Schematic representation of the plasmon-driven reduction of CO${}_{2}$. (**bottom**) Time-resolved SERS measurement indicating the intermittent nature of the signals from the physisorbed CO${}_{2}$ reactant and the catalytic product HCOOH. (**f**) Artistic representation of the nanoparticle on mirror geometry, wherein NBT molecules are sandwiched between a Ag nanoparticle and a Au thin film. (**g**) Direct and indirect reaction pathways for the reduction of NBT to ABT. (**h**) Time-resolved SERS intensities of NBT and DMAB, along with the fit to a single exponential function (blue) and step function (red). Panel (**a**) is reproduced with permission of [69]. Copyright Nature Publishing Group, 2012. Panel (**d**) is reproduced with permission of [80]. Copyright Royal Society of Chemistry, 2013. Panel (**e**) is reproduced with permission of [93]. Copryright American Chemical Society, 2018. Panels (**f**–**h**) are reproduced with permission of [84]. Copyright American Chemical Society, 2016.

**Figure 3 nanomaterials-10-02377-f003:**
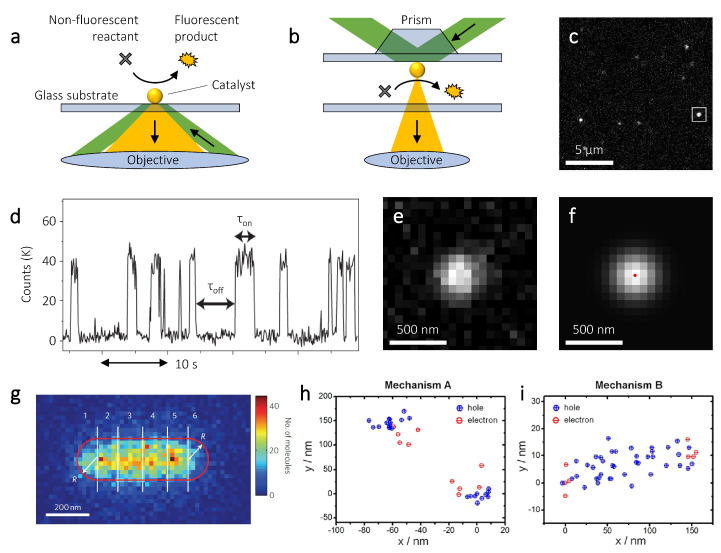
Super-resolution catalysis mapping on individual metal nanoparticles. (**a**,**b**) A non-fluorescent reactant converts to a fluorescent product, which can be imaged in an optical fluorescence microscope. Total internal reflection fluorescence (TIRF) illumination is achieved through the objective (**a**) or through a prism (**b**). (**c**) Widefield image with fluorescent events on multiple particles. (**d**) Time trace of the fluorescence intensity measured on a single catalytic particle; $\tau_{off}$ is the product formation time and $\tau_{on}$ is the product desorption time. (**e**,**f**) A single fluorescent burst (**e**), corresponding to a single reaction product, can be fit to a two-dimensional Gaussian (**f**), which results in the position of the molecule (red dot in panel f). (**g**) Two-dimensional histogram of resorufin molecules detected on a single gold nanorod. The nanorod is divided in 6 segments and for each segment a specific turnover rate can be calculated. (**h**,**i**) Super-resolved locations of resorufin molecules detected on a single CdS-Au nanorod. Blue points correspond to hole-driven reactions and red points to electron-driven reaction. The nanorod is excited below bandgap with a 532 nm laser (**h**) or above bandgap with a 405 nm laser (**i**). Panels (**c**,**e**,**f**) are reproduced with permission of [110]. Copyright American Chemical Society, 2019. Panel (**d**) is reproduced with permission of [29]. Copyright Nature Publishing Group, 2008. Panel (**g**) is reproduced with permission of [31]. Copyright Nature Publishing Group, 2012. Panels (**h**,**i**) are reproduced with permission of [114]. Copyright American Chemical Society, 2014.

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
