# Peer review of "Single Particle Approaches to Plasmon-Driven Catalysis"

_nanomaterials, 2020, doi:10.3390/nano10122377_

Round 1
Reviewer 1 Report
This manuscript described the use of SERS and super-resolution fluorescence techniques to study plasmonic catalysis at a single particle level. The contents are relevant and well-organized. I recommend the acceptance of this manuscript after minor revisions. My specific comments are:
1) The title should be changed to be more relevant to their discussion. While the title mentioned about "energy", the only example that involves "energy conversion" is on CO2 reduction. However, bulk of the discussion is on the conversion of dye molecules and the dimerization of NPT. Moreover, this manuscript is more on the plasmonic reaction investigation rather than "single particle approaches to energy conversion".
2) While super-resolution fluorescence microscopy can study heterogenity in a single plasmonic catalyst, I am unsure what additional advantages can be achieved when using SERS to study single particle catalysis. The authors could provide information on the additional insight that can be gained when performing single-particle SERS compared to investigating on particle aggregates.
3) Could the authors also compare and contrast between the two optical techniques?
Author Response
Please see our reply in the attachment

Reviewer 2 Report
Very interesting paper and suitable for publication. Some points are explanatory of the properties of the localized plasmons and the Raman effect and can be explanatory for the reader and do not weigh down the reading. The approach related to two far-field optical microscopy techniques, already known, surface-enhanced Raman spectroscopy and super-resolution fluorescence microscopy, are despite everything useful interesting.
I find only a limitation linked to the size of the nanoparticles used in the paper.
It would be interesting a study that allows tuning the wavelength of the plasmonic absorption peaks to the specific application. I suggest further work that also takes into account the variable size of the nanoparticles.
Author Response
Please see our reply in the attachment

Reviewer 3 Report
This review introduces studies that have revealed catalytic reactions at the single metal nanoparticle level by SERS and fluorescence microspectroscopy. The topics are of interest to researchers in a wide range of fields. It also describes in detail the spectral characteristics of plasmons, their theory, and the characteristics of SERS and fluorescence spectroscopy. The references are also well cited and therefore this review might be published in Nanomaterials after a minor revision.
- The introduction states that it is important to elucidate for single-particle level catalytic reactions using dark-field optical arrangements, etc. Therefore, when introducing an experiment on a rough surface of silver, the authors are recommended to explain why the study is important.
- As the author described, high spatial resolution experiments at the single nanoparticle level are important in elucidating the mechanism. On the other hand, time-resolved measurement with high temporal resolution of femtoseconds and picoseconds such as plasmon-induced electron transfer process and reaction intermediates is also considered important for understanding the mechanism. The authors should comment on that.
- The authors do not introduce measurements with scanning near-field optical microscopes or other near-field microscopes. Research on catalytic reactions using a near-field microscope is also important for high spatial resolution measurements. The authors are recommended to describe them as well.
Author Response
Please see our reply in the attachment
